# Temporally and Spatially Resolved Emission Spectroscopy of Hydrogen, Cyanide and Carbon in Laser-Induced Plasma

**Christian G. Parigger** [1,*] **, Christopher M. Helstern** [1] **and Ghaneshwar Gautam** [2]

[1] Department of Physics and Astronomy, University of Tennessee/University of Tennessee Space Institute, 411 B. H. Goethert Parkway, Tullahoma, TN 37388, USA

[2] Fort Peck Community College, 605 Indian Avenue, Poplar, MT 59255, USA

[*] Correspondence: cparigge@tennessee.edu; Tel.: +1-931-841-5690

**Abstract:** In this study, we examine the atomic and molecular signatures in laser-induced plasma. Abel inversions of measured line-of-sight data reveal insight into the radial plasma distribution. Laser-plasma is generated with 6 ns, Q-switched Nd:YAG radiation with energies in the range of 100 to 800 mJ. Temporally- and spatially-resolved emission spectroscopy investigates expansion dynamics. Specific interests include atomic hydrogen (H) and cyanide (CN). Atomic hydrogen spectra indicate axisymmetric shell structures and isentropic expansion of the plasma kernel. The recombination radiation of CN emanates within the first 100 nanoseconds for laser-induced breakdown in a 1:1 mole ratio $CO_2$:$N_2$ gas mixture. CN excitation temperatures are determined from fitting recorded and computed spectra. Chemical equilibrium mole fractions of CN are computed for air and the $CO_2$:$N_2$ gas mixture. Measurements utilize a 0.64-m Czerny–Turner type spectrometer and an intensified charge-coupled device.

**Keywords:** laser-induced plasma; atomic spectroscopy; molecular spectroscopy; cyanide; hydrogen; carbon

## 1. Introduction

Analysis of atomic species traditionally utilizes atomic emission spectroscopy of flames, plasmas, arcs, or sparks to quantify elements in the sample. However, elemental composition studies frequently apply laser-induced breakdown spectroscopy (LIBS) in a variety of environments and with extensions to molecular characterizations of solids, liquids, and gases [1–4]. Investigations of laser-induced hydrogen plasmas serve the purpose of addressing fundamental aspects of time-resolved emission spectroscopy and associated dynamic processes following optical breakdown. Measurements of hydrogen plasmas allow one to determine important plasma parameters such as excitation electron density and temperature. Usually, one measures the width of Balmer series lines of hydrogen that occur in the visible spectrum for electron density determination, and one integrates the area of these lines with respect to the continuum and constructs Boltzmann plots for excitation temperature inferences. Optical emission spectroscopy (OES) indicates the presence of molecular species that can elucidate sample composition or interaction processes in the ambient atmosphere [3–6]. Typically, atomic hydrogen lines, $C_2$ Swan bands, and the cyanide (CN) violet system appear in various applications [6,7] of laser-induced OES such as in experiments with hydrocarbons [8].

Applications of temporally and spatially resolved emission spectroscopy include laser ablation molecular isotope spectrometry [9], combustion analyses [10], and plant or medical diagnostics [11–13]. Detection methods of CN in the medical field include optical methods, electrochemical methods, mass spectrometry, gas chromatography, and quartz crystal mass monitors [14]. These methods are

adequate for the detection of cyanide, yet molecular emission spectroscopy allows one to measure the composition with a minimally invasive approach.

In this study, we report selected results from experiments using nanosecond laser spectroscopy of hydrogen plasma [15] and of CN molecular emission spectroscopy [16]. The CN recombination radiation occurs within the first 100 nanoseconds for laser-induced breakdown in 1:1 mole ratio $CO_2$:$N_2$ gas mixtures at a pressure of 1 atm. Aspects of the analysis include Abel inversions [17–21] and computational modeling [6] of the plasma. Analysis of asymmetric plasma expansion would require Radon inverse transformations [22]. Use of a chemical equilibrium code [23] predicts species distribution as a function of temperature and pressure, and these results are used in the investigation of the expanding laser-plasma.

## 2. Experimental Details

The experimental arrangement consists of a set of components typical for time-resolved, laser-induced optical emission spectroscopy [14,15], or nanosecond laser-induced breakdown spectroscopy (LIBS). Primary instrumentations include a Q-switched Nd:YAG device (Quantel model Q-smart 850) that is operated at the fundamental wavelength of 1064-nm to produce full-width-at-half-maximum 6-ns laser radiation with an energy of 850 mJ per pulse, a laboratory type Czerny–Turner spectrometer (Jobin Yvon model HR 640) with a 0.64-m focal length and equipped with a 1200 grooves/mm grating, an intensified charge coupled device (Andor Technology model iStar DH334T-25U-03) for recording of temporally and spatially resolved spectral data, a laboratory chamber or cell with inlet and outlet ports together with a vacuum system, electronic components for synchronization, and various optical elements for beam shaping, steering, and focusing.

An initial experimental study explores optical breakdown in laboratory air at standard ambient pressure and temperature (SATP). The 850 mJ/pulse beam from the laser device is expanded to a diameter of 1 cm. For the shadow-graph visualizations, a fused silica plano-convex lens (Thorlabs model LA4545) focuses the laser beam with f/10 optics. The f-number (f/#) is computed as the ratio of beam focal length and the beam diameter at the lens. Figure 1 illustrates computed radial distributions [6] for the Thorlabs LA4545 lens for focusing with f/5 and f/10 optics. The peak irradiance distributions are computed for 850 mJ, 6 ns, 1064 nm radiation. The tighter f/5 focusing reveals about one order of magnitude (or by a factor of $2^3$) smaller focal volume than that obtained for f/10 focusing. The optical breakdown thresholds at 1 atm in dry air for 1064-nm radiation is 0.28 TW/cm$^2$ [24,25]. The cyan or light-blue pseudo-color and above show irradiance levels 3 to 10 times above breakdown threshold. There are several bead-like spots that promote initiation of optical breakdown along the optical axis.

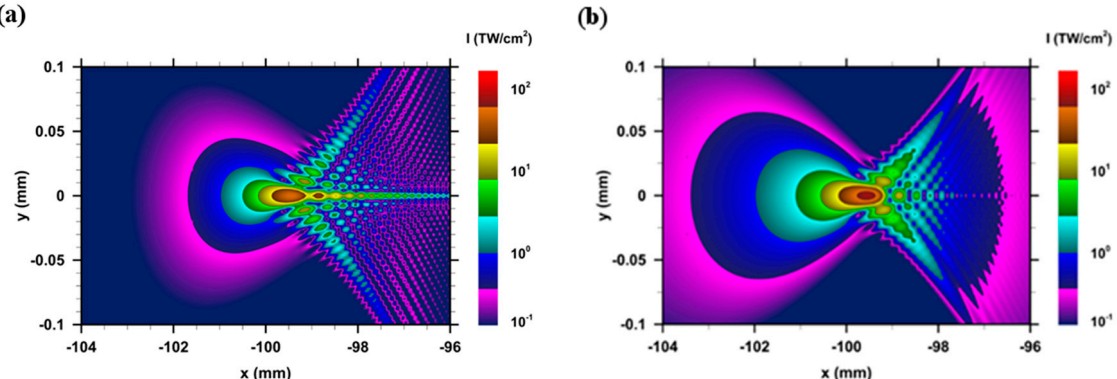

**Figure 1.** Spatial distribution of (**a**) f/5, and (**b**) f/10, 1064-nm focusing with the Thorlabs LA4545, 100-mm focal length lens [15].

For the CN experiments, a singlet lens (Thorlabs model LA1509-C) accomplishes with f/5 focusing the generation of the optical breakdown micro-plasma in a chamber that contains the 1:1 mole ratio $CO_2$:$N_2$ gas mixture at 1 atm (Airgas ultrahigh purity $N_2$ and research grade $CO_2$). Beam-splitters and apertures attenuate the energy/pulse from 850 mJ to 150 mJ for the CN experiments. The reduction in energy per pulse from 850 mJ to 150 mJ implies that the yellow regions in Figure 1 indicate irradiances about 3 times above threshold. In view of the measurements of breakdown spectra, the slit of the monochromator is parallel to the beam path, and the slit dimension is parallel to the x-dimension in Figure 1.

Captured shadow-graphs of the breakdown plasma serve the purpose of visualizing the plasma expansion [26]. Figure 2 illustrates typical shadowgraphs recorded in SATP laboratory air. The figures reveal vertical stagnation layers that originate from multiple breakdown sites. Multiple breakdown sites are indicated in Figure 1 by the bead-like focal intensity distribution along the optical axis. The irradiance in the CN experiments is about 1 order of magnitude smaller than in the air-breakdown visualizations, therefore, the number of breakdown spots along the optical axis is reduced and the apparent shape of the breakdown region would become closer to spherical symmetry.

**(a)** 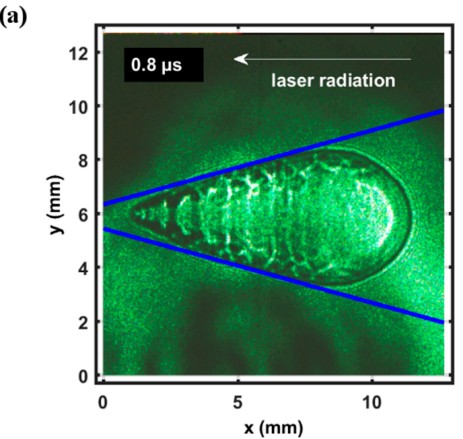 **(b)** 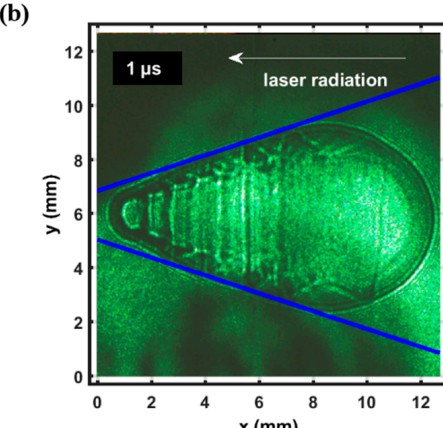

**Figure 2.** Shadowgraphs (**a**) 0.8 μs and (**b**) 1 μs. At 1 μs, the plasma expands vertically at ~Mach 3 (~1 km/s). The slopes in (**a**) and (**b**) are ±3.6 and ±3, respectively, indicate the forward shock-wave envelopes [26].

The laser-supported plasma expansion is consistent with previous focal volume investigations [27] and with the Taylor–Sedov blast-wave model [28,29]. Increased electron density and temperature occur in the outer region of the plasma kernel as evidenced by the bright-to-dark boundaries that appear to cause multiple reflections inside the shock wave. The vertical extend is about a factor of 1.4 smaller for 150 mJ pulses than that for 850 mJ pulses, according to the Taylor–Sedov energy dependency for the radius of a spherical expansion.

## 3. Results and Discussion

The experimental series for the separate measurements of atomic H and CN molecular distribution after optical breakdown includes evacuating the cell with a mercury pump to a pressure of $10^{-4}$ Pa ($10^{-6}$ Torr), and then introducing hydrogen or the $CO_2$:$N_2$ mixture. Figure 3 illustrates typical raw images of the captured time-resolved data following optical breakdown [16] in the ultrahigh pure $N_2$ and research grade $CO_2$.

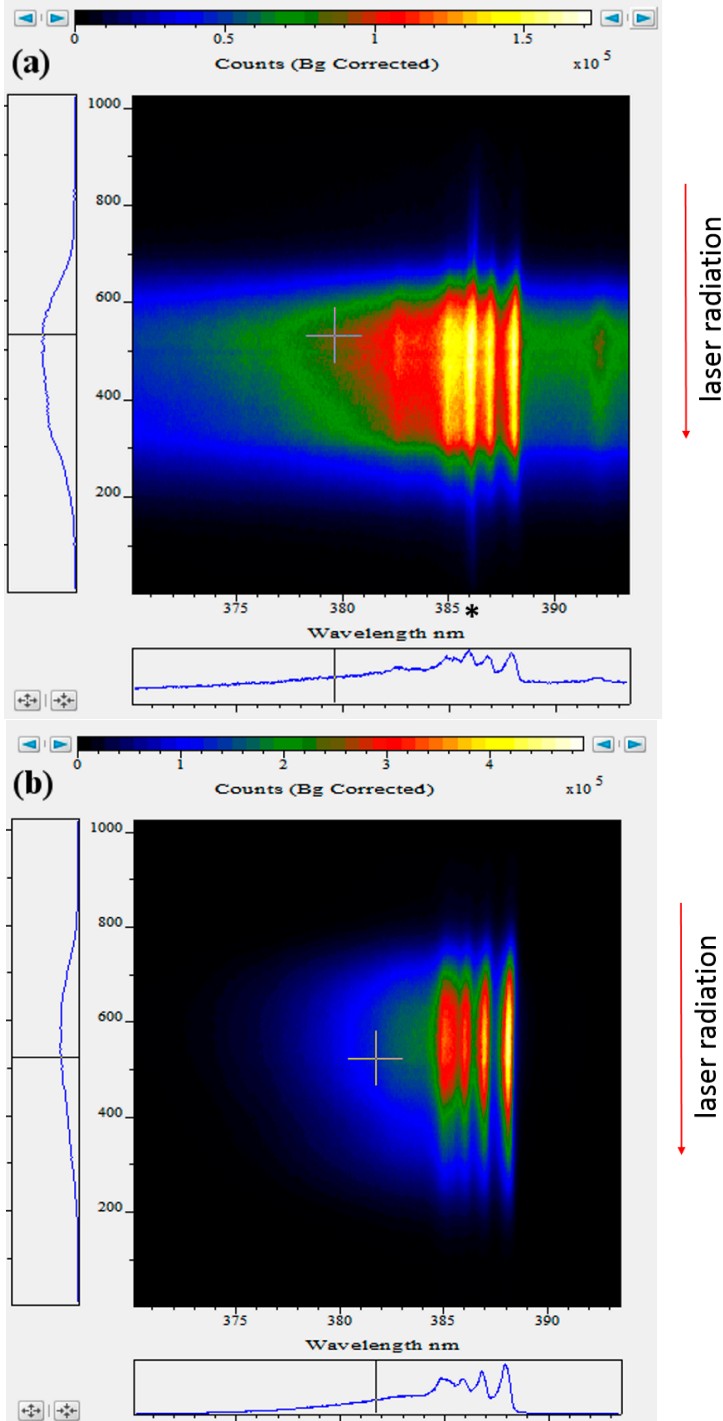

**Figure 3.** Raw spectra (**a**) 0.45 µs and (**b**) 3.7 µs after optical breakdown in a 1:1 $CO_2$:$N_2$ gas mixture at 1 atm. The asterisk near 386.2 nm indicates in (**a**) the carbon line measured in 2nd order [15]. This carbon line dominates the spectra in (**a**) near the edges at pixel heights of 200 and 600, but (**b**) shows only well-developed CN spectra.

In the reported investigations, signatures of the 0-0, 1-1, 2-2, 3-3, and 4-4 band heads begin to emanate for time delays of the order of 0.1 µs from optical breakdown. Moreover, the plasma typically propagates towards the laser side. The recorded data indicate a ~0.8 mm CN-signal propagation in the 370 nm to 393.5 nm spectral, 7 mm object window during the first 5 µs, from a delay of 0.2 µs to 5.2 µs. Optical breakdown inside the chamber occurs at a rate of 10 Hz, with the laser beam focused with f/5

optics from the top, or parallel to the slit. The detector pixels are binned in four tracks along the slit direction, resulting in obtaining 256 spectra for each time delay. Figure 3 shows accumulated raw data from 100 consecutive optical breakdown events, recorded at a time delay of 0.450 µs and a gate width of 0.125 µs. The vertical axis indicates the slit height, the laser beam is focused from the top.

However, a Czerny–Turner monochromator has no strict stigmatic imaging of slit height to position on the ICCD array, thus the spectra along the laser path are approximately related to the position. With 1:2 imaging, and a pixel resolution of 13.6 µm, the discernable plasma size in the cell amounts to ~3 mm. The figure illustrates that the CN band heads of the Δv = 0 sequence are well-developed, and it also displays an atomic line near 386.2 nm that is the carbon C I 193.09-nm atomic line, recorded in second order [16].

Abel inverse transformation allows one to obtain the radial distribution of the plasma. For Abel inversion, line-of-sight data of radially symmetric profiles are required. Analysis of the molecular CN spectra utilizes the same methods as previously applied for analysis of atomic hydrogen spectra [15,16]. Moreover, the results for the CN spatial distribution are expected to show expansion phenomena that are analogous to those elaborated for hydrogen laser-plasma [15]. The integral equation describes line-of-sight averaging,

$$I(z, \lambda) = 2 \int_z^\rho I(r, \lambda) \frac{r}{\sqrt{r^2 - z^2}} dr. \tag{1}$$

The measured line-of-sight data, $I(z, \lambda)$, along the slit dimension, $z$, are inverted for each wavelength, $\lambda$, to obtain the volumetric radial distribution, $I(r, \lambda)$, with the upper limit much larger, $\rho \gg R$, than the radius, R, of the plasma. The choice of the number of Chebyshev polynomials for the inversion [17] is equivalent to the use of a digital filter [18] that causes broadening of computed radial spectra. In this work, the inversion uses 10 polynomials, a smaller number of polynomials would cause smaller spectral resolution.

Figure 4 displays results of the Abel-inverted hydrogen data for a time delay of 0.4 µs from optical breakdown. For the recorded data, the constructed Boltzmann plots utilize $H_\alpha$, $H_\beta$, and $H_\gamma$ integrated line shapes to provide a measure for the excitation temperature distribution [15]. Figure 4 portrays a cooler central region and a relatively hot ring of the order of 100,000 K (8.6 eV). As indicated in the figure, the kernel expands at or near the speed of sound in hydrogen gas.

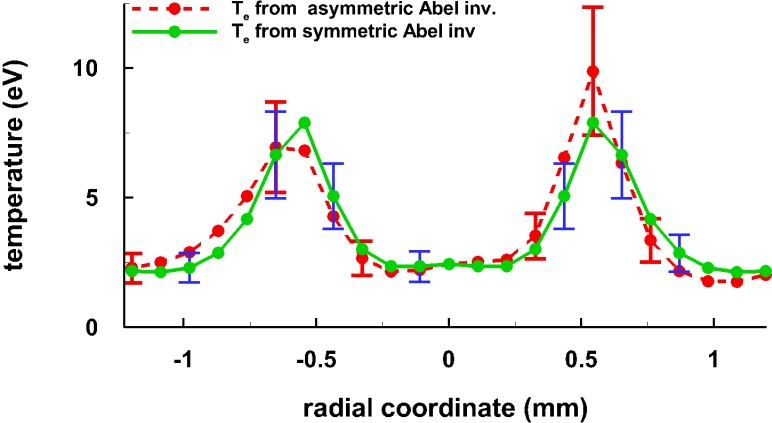

**Figure 4.** Electron temperature, $T_e$, vs. radial position [15].

Figure 5 shows the corresponding electron densities [15,19,20]. The error bars are in part due to the lower fidelity of the Abel inversion for a time delay of 0.4 µs.

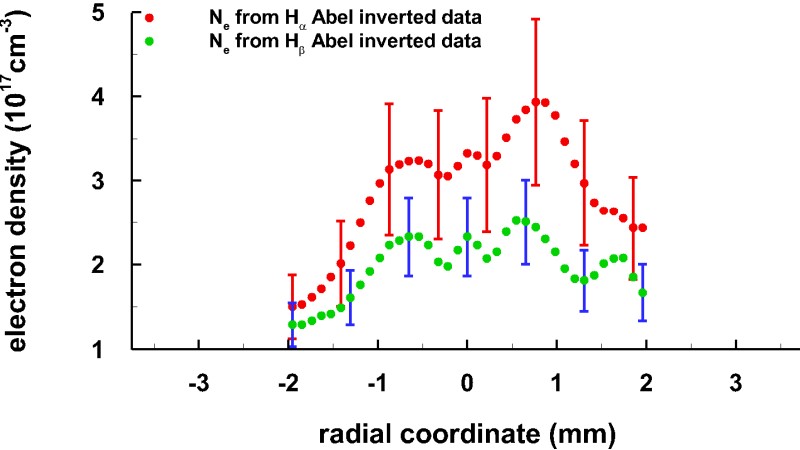

**Figure 5.** Electron density, $n_e$, vs. radial position [15].

Wavelength and detector system-sensitivity calibrated CN spectral data are Abel-inverted [16]. Figure 6 displays the results. Analogous to recently reported hydrogen nitrogen gaseous mixtures [30], the CN signals are weaker at the center and indicate a slightly lower temperature than at 0.85 mm. There appears to be residual interference from an atomic line at 386.2 nm that indicates the carbon line at the 193.09-nm line [20,21], measured in second order.

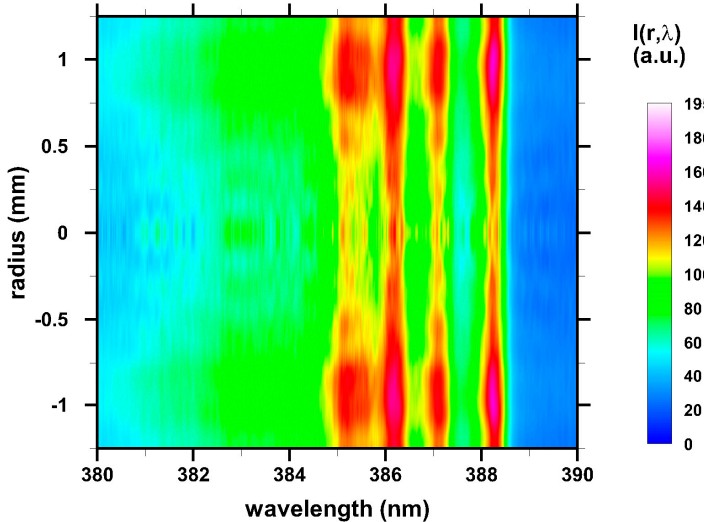

**Figure 6.** Abel-inverted spectra versus radius at 1.2-µs time delay, gate 0.125 µs [16].

Figure 7 illustrates measured and fitted CN spectra at the center and at a radius of 0.85 mm. A modified Boltzmann plot method [31] is utilized in the recently published program [32] for the determination of the best-fit FWHM, $\Delta\lambda$, and temperature, T, from the entire spectrum. The determination of the temperature and mole fraction of CN and carbon line-width along the slit is topic of ongoing research. The variation is expected to be analogous to recent hydrogen–nitrogen laser-plasma work [33]. Recorded spectra at a time delay of 3.7 µs and at 1 mm show lower temperature [16] than that reported here for the 1.2 µs time delay.

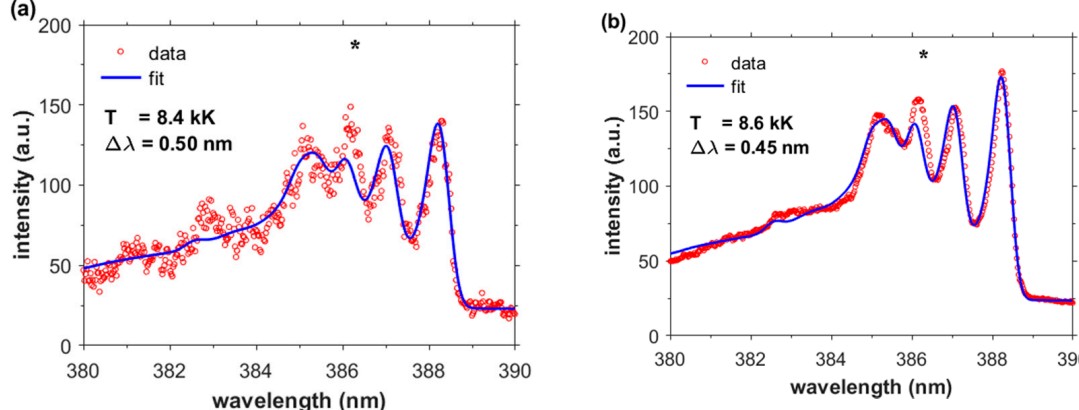

**Figure 7.** Inferred CN excitation temperature (**a**) at the center, (**b**) at a radius of 0.85 mm, 1.2-μs time delay, gate 0.125 μs. The asterisks indicate the 386.2-nm position for the 193.1-nm carbon line measured in 2nd order.

One would expect that the CN molecule distribution is close to uniform in chemical equilibrium. For time delays in the range of 5 μs to 50 μs, the line-of-sight molecular CN spectra are well-developed, and the recorded optical emissions originate from a decreasing volume with increasing gate delay. In addition, CN recombination radiation signals may be stronger for specific ranges of temperature. Computation of the freely available Chemical Equilibrium with Applications (CEA) code [23] elucidate CN mole fractions versus temperature. Several atoms and molecules including ionic species are part of the CEA computations, but the results for the CN mole fractions are of primary interest in this work. Figure 8 shows the CN distribution for both air and the mixture as function of temperature.

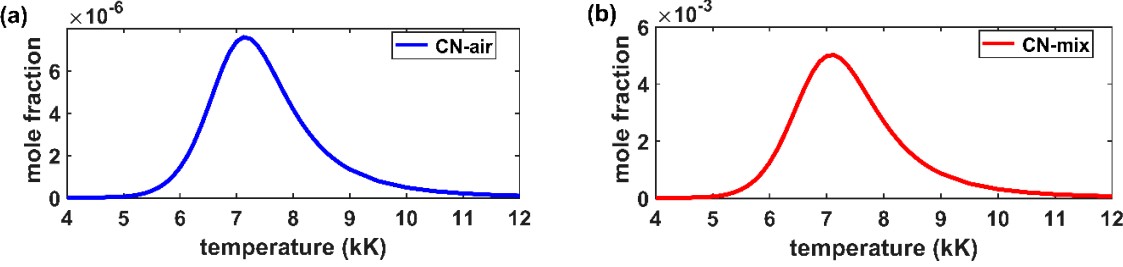

**Figure 8.** CN (**a**) air and (**b**) mixture mole fraction vs. temperature in chemical equilibrium, CN shows a maximum near 7 kK for the 1:1 mole ratio $CO_2$:$N_2$ mixture at 1 atm. CN fractions in air are nearly three orders of magnitude lower than that for the mixture.

The recombination signals from the mixture are strongest near 7 kK. Using the tabulated CEA results [23] at a temperature of 7 kK, one finds that the number of CN molecules in the mixture and air amount to mole fractions of and $\sim 5 \times 10^{-3}$ and $\sim 8 \times 10^{-6}$, respectively. The CEA program computes a volley of species that are included for completeness: e-, C, $C^+$, $C^-$, CN, CN+, $CN_-$, CNN, CO, CO+, $CO_2$, $CO_2^+$, $C_2$, $C_2^+$, $C_2^-$, CCN, CNC, $C_2O$, $C_3$, N, $N^+$, $N^-$, NCO, NO, $NO^+$, $NO_2$, $N_2$, $N_2^+$, $N_2^-$, NCN, $N_2O$, $N_3$, O, $O^+$, $O^-$, $O_2$, $O_2^+$.

The spectra analyses of the central region data at 0.45-μs time delay indicate slightly lower temperatures at center than that obtained at a radial position of 0.85 mm, and spectroscopic interference from the carbon line in 2nd order is apparent for time delays of 0.45 μs and 1.2 μs. Line-of-sight data consist of average spectra with contributions from regions at different temperatures. For instance, the molecular CN spectra near the plasma edges, viz. near the top and bottom of the spectra (see Figure 3), show smaller signals but reveal higher temperatures than in the center portion.

## 4. Conclusions

Measured hydrogen and cyanide recombination spectra indicate a spherical shell structure of the plasma kernel inside the shock wave early in the plasma decay. For hydrogen, electron density and excitation temperature are higher in the peripheral region than near the center due to expansion dynamics of the plasma kernel after optical breakdown.

The recombination radiation from CN shows a similar trend for early time delays, namely, higher excitation temperatures occur in the outer region than near the center of the plasma kernel. Occurrence of this trend is also supported by the carbon line that overlaps with the CN spectra near the 2-2 vibrational transition. The application of Abel inversion requires a symmetric light source, yet the analysis of the extent of asymmetry in the hydrogen plasma leads to variations within the estimated error bars.

Shadow-graph studies in air would support symmetrizing the spectral data recorded in the $CO_2$:$N_2$ mixture and applying Abel inverse transforms for extraction of the spatial variation of the CN optical emission signals. CN plasma characterization will improve current CN detection methods and applications of the presented work extend to analysis of supersonic laser-plasma expansion, transient laser-induced chemistry, to name two examples.

**Author Contributions:** C.G.P. and G.G. conceived and performed the experiments and C.G.P. and C.M.H. analyzed the result. All authors contributed to writing the article.

**Funding:** The authors appreciate the support in part by the Center for Laser Application, a State of Tennessee funded Accomplished Center of Excellence at the University of Tennessee Space Institute.

**Conflicts of Interest:** The authors declare no conflict of interest.

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
