# Peer review of "Temporally and Spatially Resolved Emission Spectroscopy of Hydrogen, Cyanide and Carbon in Laser-Induced Plasma"

_atoms, doi:10.3390/atoms7030074_

Round 1
Reviewer 1 Report
Review on “Temporally and spatially resolved emission spectroscopy of cyanide, hydrogen and carbon in laser-induced plasma”
by Christian G. Parigger, Christopher M. Helstern, and Ghaneshwar Gautam
The paper addresses some issues of laser induced breakdown in common gases H2, CO2/N2, and air. The message of this manuscript is quite unclear: it consists of fragments which describe the works done by the authors before and weakly linked to each other. Many results are lacking proper explanations and I do not see a particular novelty in this material. I was unable to access some references (e.g. Int. Rev. At. Mol. Phys.) but note that at least some of the results have been published before.
Here are the specific comments.
Abstract is inaccurate as it claims “Investigations include solids, gases, and nano-particles.” The paper contains no ablation on solids and no study of nanoparticles. Neither can I see how “expansion dynamics and turbulence due to shock phenomena are elucidated to address local equilibrium details”. Fig.2 shows two shadowgraphs that are not related to images and spectra shown later in the paper.
The phrase “Experiments with femtosecond laser pulses reveal…” (p.2, top) is out of context as no experiment with fs laser is offered and no comparison with current ns results is made.
Speaking about “1:1 CO2:N2 atmospheric (pressure-?: Rev.) gas mixture…” (p.2) the authors should specify whether it’s the mass or mole ratio.
The authors make a link between Fig.2 and Fig.3 saying “The figures reveal vertical stagnation layers that originate from multiple breakdown sites as indicated in the computed focal intensity distribution”. It would be instructive to discuss how/weather the train of micro-breakdowns is formed how it is linked to the multi modal distribution shown in Fig.1. The authors should provide a threshold for air breakdown and relate that to the intensities of nodes shown in Fig.1.
What is the “…nominal mercury pump vacuum …” (p.3)?
The geometry of experiment for performing the Abel inversion does not look correct “…with the laser beam focused with f/5 optics from the top, or parallel to the slit” (p.4) unless the plasma has a spherical shape. Given a tear-like shape as in Fig.2, the sphericity assumption is doubtful. The authors try to somehow account for plasma asymmetry (this follows from the profiles given in Fig.4 and assessment in Conclusions about “spherical shell structure of the plasma kernel”), however, they provide no explanation of how they do that. The explanation should be given.
Figures 4 and 5 are reproduced from Ref. [15]; the authors should clearly say that (e.g. in figure captions) and also add a label for x-axis in Fig.4. If other figures in this manuscript have also been published before, the authors should also say that.
A sentence related to Fig.6 “Figure 6 displays the results and it shows a comparison of computed and of measured spectra” (p.5) is misleading as this figure does not display any comparison. The Abel-inverted spectrum looks symmetrical; the authors should explain how they symmetrized the image before the Abel inversion.
Concerning Fig.7, I suggest the authors mark the vibrational bands, i.e. (0, 0), (1, 1), etc. and indicate (a) which software was used for fitting the bands; (b) which band the FWHM of 0.45 nm is attributed to, and (c) the origin of the temperature 8600 K (from fitting?). The 0.85 mm radius, where the molecular spectrum is seen, corresponds to plasma periphery; can the authors provide the value for T in plasma center for the completeness of physical picture?
The equilibrium concentration of CN as a function of T is incomplete without giving the concentrations of other species involved in equilibrium calculations. If not showing in the figure, the authors should at least provide a list of molecules, atoms, ions, and anions which were involved in the equilibrium calculations. The result will strongly depend on whether or not all the important species were taken into account.
Finally, I suggest the authors underline a practical value of their investigation in the conclusion section.
Even though the novelty of this manuscript seems modest to me, I recommend it for publication after the major revision.

Author Response
Dear Reviewer,
Thank you very much for your comments. In response to your comments, we edited the document and indicate the changes in green text.
(a) Abstract: Shortened the abstract for consistency with "claims"
(b) Removed the last sentence in introduction to avoid explicitly stating with femtosecond pulses, but leave the reference to femtosecond work by others (e.g., Refs. [10, 12, 13])
(c) In the Experimental details, a clear distinction is now made that indeed the shadowgraph experiments are conducted in air and with 850 mJ/pulse.
(d) Several explanatory sentences are included regarding interpretation of the 850 mJ/pulse air breakdown and 150 mJ/pulse CN experiments.
(e) The phrase “nominal mercury pump vacuum” is edited, and reference is made to 'mole ratio.'
(f) An extra figure regarding the temperature variation across the plasma is added, and an extra reference is included (Ref. [33]).
(g) A list of species used in the calculation is included.
(h) In the conclusions, a summary sentence is added (reference to applications are also in the introduction)
(i) References are included for figures as well, in addition to the existing references in the text. The referenced publications for the figures are "open access" papers, of course, we are happy to cite the reference in the figure-caption as well.
(j) The fitting program determines the best fit temperature and delta-lambda, a comment is added.
Respectfully,
Reviewer 2 Report
ATOMS 545150
Temporally and ... laser-induced plasma, Parigger et al.
The paper reports detection of CN in a laser-induced plasma. The results sound to be reliable and can be published. However, some changes must be made.
The paper is a good example for a text written by researchers which are quite familiar with experiment and data evaluation and which cannot imagine that other people need more information.
Abstract: here "background contributions" are mentioned. Nevertheless, I cannot find mentioned such contributions in the text.
1. Introduction, 1st paragraph.
- here a ".. integrated ratio of these lines .." is mentioned. It is dot said if an intensity ratio etc is meant.
2. Experimental details, 2nd paragraph
- what is meant by "1:1 CO2 N2 atmospheric gas mixture" ? A gas consisting of 50% CO2 and 50% N2 and nothing else, or mixture of this with atmospheric air (in Fig.8 also results from measurements in air are shown) ?
- Fig.1 and text are not compatible. Is Fig. 1a the focusing with lens LA 1509-C ? Earlier in text f/5 LA 1509 is mentioned, later in text LA4545 f/5, but in all cases no focus length. Is the ratio f/x given by the lens or by an additional aperture?
- Fig.2. I suggest to left rotate the figures by 90°. Then the laser light comes in vertical direction, and x is the dimension along the laser path. Use in both figures the same x tics.
- Fig.3. It must be mentioned earlier that the slit of the monochromator is parallel to the laser path and thus the vertical axis in Fig. 3 corresponds roughly to the x-position in Fig.2 (mention also that a Cz.-T. monochromator has no strict stigmatic imaging of slit height to position on the CCD array, thus the spectra along the laser path are only approximatively related to the position).
-Fig.3 The spectra shown below the CCD images are too small. Mark also the spectral position 386.2 nm.
4. Results
- text "Fig. 6 displays ... a comparison of computed and measured spectra" . Disagreement with Fig. 6. Do you mean Fig. 7? If yes, Fig. 6 is not described in the text.
- in text you use unit kK, in Fig.7 not.
- Fig. 8, caption: you use again ""atmospheric CO2:N2 mixture", but in a9 measurements in air (atmosphere) and in b) results on mixture are presented.
Some remarks I made directly in the text (see appended pdf file).

Author Response
Dear Reviewer,
Thank you very much for your detailed comments. In response to your comments, we edited the document and indicate the changes in blue text.
(a) Abstract: Shortened the abstract as well in response to reviewer 1 (green text) and leave out 'background contribution' in abstract;
(b) Included edits regarding measurement of excitation temperature to clarify 'integrated ratio..'
(c) Changed atmospheric gas mixture to gas mixture at 1 atm throughout.
(d) The f/# indicates the beam diameter at the lens, added text.
(e) The paragraph regarding the different experiments and use of different lenses is edited, Fig. 3 is enlarged. However, we hope to left Figure 2 un-rotated due to the references to Figure 1. arrows are included in figure 3
(f) The comment regarding Cz.-T. is included.
(g) Figs 6 to 8 captions and references are corrected including consistent use of kK and μs
(h) Also revised the paper according to the comments in the pdf, e.g., include arrows in Fig. 3
Respectfully,
Round 2
Reviewer 1 Report
The authors did a god job in revising the manuscript; they had properly addressed all comments from the reviewer. I have no further comments and recommend this paper for publication as is.
Reviewer 2 Report
Even I find that rotation of Figs. 2 would be better, the MS seems now be ready for publication